# Thermal Evaluation by Infrared Thermography Measurement of Osteotomies Performed with Er:YAG Laser, Piezosurgery and Surgical Drill—An Animal Study

**DOI:** 10.3390/ma14113051

**Published:** 2021-06-03

**Authors:** Dragana Gabrić, Damir Aumiler, Marko Vuletić, Elizabeta Gjorgievska, Marko Blašković, Mitko Mladenov, Verica Pavlić

**Affiliations:** 1Department of Oral Surgery, School of Dental Medicine, University Hospital Center, University of Zagreb, 10000 Zagreb, Croatia; dgabric@sfzg.hr; 2Institute for Physics, 10000 Zagreb, Croatia; aumiler@ifs.hr; 3Department of Pediatric and Preventive Dentistry, Faculty of Dental Medicine, Ss. Cyril and Methodius University, 1000 Skopje, North Macedonia; elizabetag2000@yahoo.com; 4Department of Oral Surgery, Faculty of Dental Medicine, University Hospital Center, University of Rijeka, 51000 Rijeka, Croatia; marko_blaskovic@yahoo.com; 5Faculty of Natural Sciences and Mathematics, Institute of Biology, Ss. Cyril and Methodius University, 1000 Skopje, North Macedonia; m.mladenov@gmail.com; 6Department of Periodontology and Oral Medicine, Medical Faculty, University of Banja Luka, 78000 Banja Luka, Bosnia and Herzegovina; dr.vericapavlic@gmail.com

**Keywords:** Er:YAG laser, piezoelectric surgery, animal model, osteotomy, thermography

## Abstract

The bone healing process following osteotomy may vary according to the type of surgical instrumentation. The aim of the present in vivo study was to determine thermal changes of the bone tissue following osteotomies performed by Er:YAG laser ablation in contact and non-contact modes, piezoelectric surgery, and surgical drill using an infrared thermographic camera. For each measurement, the temperature before the osteotomy-baseline (Tbase) and the maximal temperature measured during osteotomy (Tmax) were determined. Mean temperature (ΔT) values were calculated for each osteotomy technique. The significance of the difference of the registered temperature between groups was assessed by the ANOVA test for repeated measures. Mean baseline temperature (Tbase) was 27.9 ± 0.3 °C for contact Er:YAG laser, 29.9 ± 0.3 °C for non-contact Er:YAG laser, 29.4 ± 0.3 °C for piezosurgery, and 28.3 ± 0.3 °C for surgical drill. Mean maximum temperature (Tmax) was 29.9 ± 0.5 °C (ΔT = 1.9 ± 0.3 °C) for contact Er:YAG laser, 79.1 ± 4.6 °C (ΔT = 49.1 ± 4.4 °C) for non-contact Er:YAG laser, 29.1 ± 0.2 °C (ΔT = −0.2 ± 0.3 °C) for piezosurgery, and 27.3 ± 0.4 °C (ΔT = −0.9 ± 0.4 °C) for surgical drill. Statistically significant temperature changes were observed for the non-contact laser. The results of the study showed beneficial effects of the osteotomy performed by the Er:YAG laser used in the contact mode of working as well as for piezosurgery, reducing the potential overheating of the bone tissue as determined by means of infrared thermography.

## 1. Introduction

Of the most frequent procedures in oral and maxillofacial surgery belong ostectomies and osteotomies. Traditional instruments such as diamond or steel burs, oscillating saws or chisels are generally used for performing this type of procedure [1,2]. Even though these instruments are considered as gold standard for bone osteotomy, they have some disadvantages such as friction and pressure, during their contact with the hard tissue, resulting in fracture of fragile bone segments. Thermal osteonecrosis is bone injury induced by frictional heat from osteotomies when the temperature arises during the osteotomies above the critical temperature of 47 °C [3,4,5,6,7,8]. The increased temperature during osteotomies correlates with cortical thickness, drill speed, sharpness, depth and force, and irrigation [3,4,5,6,7,8]. Therefore, tools for bone tissue surgery with reduced force, friction and collateral tissue trauma have become necessary. This has led to the development of new advanced and superior osteotomy techniques: piezoelectric surgery, and laser ablation [2,9]. Preparation of the dental implant site is also a type of osteotomy which is together with healthy bone a primary precursor for successful primary healing. During the drilling and trephining protocol for dental implant, mechanical and thermal damage should be minimized. Significant temperature increase in the bone can also cause tissue injury and results in unsuccessful osseointegration [3,7].

Erbium:yttrium-aluminum-garnet (Er:YAG) laser is a precise tool for bone ablation due to its wavelength of 2940 nm, that is highly absorbed by water and, to a much lower extent, by hydroxyapatite [2,10]. Er:YAG laser is a suitable tool for efficient bone ablation by micro-explosions, with little or no carbonization on the surrounding and/or underlying tissues [2,11].

Absorbed energy after the laser irradiation is converted to heat, so internal pressure occurs by vaporization of interstitial water. The many advantages of Er:YAG laser surgery are lack of vibration and metal abrasion, high detoxification and bactericidal effects, reduced tissue bleeding and adjacent tissue injury, and precise geometrical cuts with regular borders [2,10,12,13]. The Er:YAG laser is known to cause a thermally changed cortical layer of bone to a depth of ~30 μm, so this is considered harmless in the bone tissue healing process [2,10].

The novel ultrasonic technique for osteoplasty and osteotomy with a piezoelectric effect working within a frequency range of 24–32 kHz is called piezoelectric bone surgery. Its main characteristics are minimal noise production during bone ablation reducing stress among patients, requirement of less force obtained by the operator for cutting the bone in comparison with a standard rotational bur, and the fact that even in the case of accidental contact with the adjacent soft tissue, it is precisely only cutting the mineralized tissue without damaging the soft tissue as the soft and hard tissues are cut at different frequencies [2,9]. Piezoelectric bone surgery as an ultrasonic technique also offers reduced necrosis, minimal invasion of the operative site, and greater control of the device [2]. A bleed-free surgery site, enhanced healing processes, and bone regeneration are also reported to be associated with piezoelectric surgery [14]. The most significant disadvantages of piezoelectric surgery are the required precise irrigation in order to avoid overheating that causes bone devascularization, the increased operation time, and loss of vitality of the periosteum and a denaturation of alkaline phosphatase [14,15].

Infrared thermography is a well-known technique for measuring infrared energy produced from an object, converting into a radiometric thermal image and displaying a picture of distribution of surface temperature. It is a unique instrument with great ability to detect any temperature changes of the tissues, whether physiological or pathological. Thermal changes of bone tissue in dentistry are also measured with this method [16].

Currently, a limited number of investigations which compare advanced methods for bone osteotomies, such as piezoelectric and high-power laser surgery, with standard bur osteotomy in terms of bone changes is available. Therefore, the purpose of the present in vivo animal study was to determine the thermal changes of the bone tissue following osteotomies performed by piezoelectric surgery, surgical drill, and Er:YAG laser ablation in contact and non-contact modes using infrared thermography. The null hypothesis that was set for this research was that temperature changes are similar in the cases of different methods of osteotomy preparation.

## 2. Materials and Methods

The Ethics Committee of the School of Dental Medicine, University of Zagreb (0039/16) and the Animal Ethics Committee of the Faculty of Natural Sciences and Mathematics, University of Skopje approved the research protocol of this study. All experiments that were performed in the study complied with the ARRIVE guidelines and were carried out in accordance with the UK Animals (Scientific Procedures) Act, 1986 and associated guidelines, the EU Directive 2010/63/EU for animal experiments, or the National Institute of Health guide for the care and use of Laboratory animals (NIH Publications No. 8023, revised 1978). A total number of 24 Wistar rats (*Rattus norvegicus albinus*) were used. All rats were adult male and 10-week-old, each weighing 300 to 350 g and kept in specified pathogen-free conditions. There were 4 rats in each cage. Seven days before the experiment, they were allowed to acclimatize in a controlled conditions with access to water and food ad libitum at a room temperature of 22 °C, 12 h light/dark cycle.

Sample size was calculated using the G* Power package (version 3.1,University of Düsseldorf, Germany). In determination of sample size, we used a significance level of 0.05, power of test was set to 0.8, and effect size was set to 0.7.

### 2.1. Surgical Procedure

Animals were under general anesthesia by an intraperitoneal injection of thiopental sodium (Rhone-Poulenc Rorer Limited, Nenagh, Co Tipperary, Ireland), so osteotomies on both of the rat’s tibiae could be performed. Rats were positioned in dorsal decubitus and surgical sites were prepared with hair-removal and disinfection using povidone iodine solution. On each tibia a linear, 18 mm long incision was made through the skin, muscles, and periosteum. The preparation site of bone was exposed by gentle elevation and reflection of the dermo-periosteal flap. Four osteotomies were performed always in the same sequence: the right distal part of the tibia by a contact Er:YAG laser, the right proximal part by piezosurgery, the left proximal part of the tibia by low-speed surgical drill, and finally the left distal part by digitally controlled non-contact Er:YAG laser. The osteotomies were 5 mm away from each other and were approximately 2 mm deep. The final diameter of each osteotomy ranged between 1.0–2.0 mm depending on the instrument applied. During the preparations, an articulated arm delivery system of the laser, piezoelectric and low-speed handpieces were fixed to ensure adequate pressure applied during the osteotomy preparation. Biosecurity standards to protect the personnel when using the laser devices were followed. The time interval between the use of two different methods of preparation was set at 30 s in order to avoid heating of the untreated part of the bone by the previous preparation. Different times for osteotomy preparation depending on the tested method used were observed to reach similar osteotomy depths.

Contact Er:YAG laser (LightWalker^®^, Fotona, Ljubljana, Slovenia) was used with a H-14N handpiece and fiber of 1.0 mm diameter (core diameter of 940 μm) and with continuous water spray (40–60 mL/min) According to the results of the previous pilot study on the optimal contact of Er:YAG laser irradiation parameters on bone tissue healing, we set the parameters for this study as: power 7.5 W, pulse energy 375 mJ, repetition rate 20 Hz, and medium-short pulse mode (MSP mode, pulse duration 10 μs). The time required for each contact laser osteotomy was 10 s.

Non-contact Er:YAG laser (LightWalker^®^, Fotona, Ljubljana, Slovenia) with a recently developed type of circular digitally controlled handpiece (X-Runner™, Fotona, Ljubljana, Slovenia), 2 mm in diameter, with water spray set at ratio 4:4 and focal distance of 13 mm was used to perform osteotomy. The parameters were as follows: power 7.5 W, pulse energy 750 mJ, repetition rate 10 Hz, and quantum-square pulse mode (QSP, a train of five short pulses of 50 μs separated by 85 μs). The time required for each non-contact laser osteotomy was only 5 s.

Piezoelectric surgery (Piezomed, W&H Dentalwerk Burmoos GmbH, Burmoos, Austria) osteotomy was performed at maximum power with continuous water cooling using a diamond-coated spherical tip (1.2 mm diameter, S2). The average pressure applied during the instrument handling was approximately 15 N. The time required for each piezoelectric osteotomy was 30 s.

Low-speed surgical drill (Meisinger, Neuss, Germany) osteotomy was performed using a handpiece of a surgical physio dispenser at a rate of 1200 rpm and a new round steel surgical bur with 2-mm-wide diameter under constant saline irrigation. The time required for each drill osteotomy was 20 s.

All osteotomies were performed under continuous irrigation and adequate cooling to avoid thermal damage of the bone tissue during preparations. At the end of the surgery, all incisions were closed with interrupted absorbable sutures (Vicryl, 6.0; Ethicon Inc., Somerville, NJ, USA).

After surgical procedures, the rats were treated with veterinary acetaminophen 1 mg/kg in 1 L of water and kept under the same laboratory conditions for an additional one week. The animals that failed to recover from the procedure, in terms of inability to move and walk, were euthanized with an overdose of the previously administered anesthetic solution.

### 2.2. Thermographic Procedure

Measurements of temperature change during the four different osteotomies were performed using an infrared (IR) thermographic camera (FLIR T335; FLIR Systems Pty Ltd., Melbourne, Australia) with a detection range of −20 °C to +650 °C, a thermal sensitivity of <50 mK, and IR resolution of 320 × 240 pixels. Analysis of the recorded data was made using the FLIR Tools software (FLIR Systems Inc., North Billerica, MA, USA). Prior to the surgical procedure the camera was set up on a tripod at a fixed distance (30 cm from the rat’s tibia) and the settings were always the same. Before imaging, the animals were left in the recording room for a minimum of 10 min. Operating and recording room had the same controlled conditions such as humidity and temperature for the whole time. Imaging was started 5 min before the start of the single osteotomy preparation and was completed 5 min after the end of the preparation. Temperature measurements were recorded at every second during osteotomy preparation. The thermal camera recorded a separate movie for each osteotomy. Movie frames were then extracted as thermal camera images, and the position of the osteotomy was determined on the images (frames) by visual inspection and marked as a region of interest (ROI) (Figure 1). All recorded movie frames were read-out for this point, thus providing the information on the temperature at the ROI during the performance of osteotomy. In this way we measured the time evolution of the bone temperature at the point of osteotomy. Baseline temperature and maximum temperature values were obtained from previously recorded movie frames for each surgical technique on each individual experimental animal. Baseline temperature (Tbase) was the initial recorded temperature before osteotomy preparation, maximum temperature (Tmax) was the highest temperature recorded during osteotomy preparation, and mean temperature (ΔT) was calculated for each measurement. Only temperature results that could be clearly read from all measurement points in the area of ROI were used. Values of the temperature at which the instrument holder or fixative partially covered the surface of the bone on which the value was measured were excluded from the sample.

### 2.3. Statistical Analysis

Statistical analysis was performed using the SAS statistical package (SAS 9, SAS Institute, Cary, NC, USA) on the Windows platform. Descriptive parameters (mean values and standard errors) were calculated. The difference for the registered temperature changes between groups was assessed by the ANOVA test for repeated measures with the Greenhouse–Geisser correction. Compliance of data with the test requirements was assessed by Shapiro–Wilk’s test and Mauchly’s sphericity test. For ‘post-hoc’ comparison, the paired t-test with Bonferroni correction was used. The level of significance was set at 0.05.

## 3. Results

In the first step, the normality of the data was tested. For all methods, except piezosurgery the distribution does not deviate from the normal distribution (Table 1, *p* > 0.05; Shapiro–Wilk test). The distribution for piezosurgery deviated slightly from the normal distribution (*p* = 0.02), so the application of the ANOVA test was justified (since it was actually four tests, the limit value could be set to 0.05/4 = 0.0125, so even for this method we did not deviate from the normal distribution).

The number of analyzed rats was reduced (N-15) because all the results that were insufficiently precise according to the measured points of ROI were not considered for further analysis.

There was a significant difference in temperature change between all techniques (Table 2, *p* < 0.0001; ANOVA test). A *t*-test for paired samples was used to determine techniques that differed from each other. Multiple comparisons showed that a difference exists between all techniques except between piezosurgery and surgical drill. The largest temperature change was observed for the non-contact laser (53.3 °C), followed by the contact laser (2.0 °C) and the smallest for piezosurgery and surgical drill (−0.0 °C and −1.1 °C, respectively).

### 3.1. Baseline Temperature

For each measurement, the temperature before the osteotomy, baseline temperature (Tbase), and maximum temperature were measured, and ΔT = Tmax – Tbase was then calculated for each measurement. Measurements during the osteotomy (Tbase, Tmax) were determined for each surgical technique (Figure 2, Figure 3, Figure 4 and Figure 5). Mean baseline temperature (Tbase) was 27.9 ± 0.3 °C for contact Er:YAG laser, 29.9 ± 0.3 °C for non-contact Er:YAG laser, 29.4 ± 0.3 °C for piezosurgery and 28.3 ± 0.3°C for surgical drill.

### 3.2. Maximum Temperature

Mean maximum temperature (Tmax) was 29.9 ± 0.5 °C (ΔT = 1.9 ± 0.3 °C) for contact Er:YAG laser, 79.1 ± 4.6 °C (ΔT = 49.1 ± 4.4 °C) for non-contact Er:YAG laser, 29.1 ± 0.2 °C (ΔT = −0.2 ± 0.3 °C) for piezosurgery, and 27.3 ± 0.4 °C (ΔT = −0.9 ± 0.4 °C) for surgical drill (Table 3).

### 3.3. Temperature Changes

A typical temperature measurement for the contact Er:YAG laser first showed several seconds of constant temperature at about 28–29 °C. This was the temperature before the laser was switched on, i.e., the baseline temperature (Tbase). After the laser was on, the temperature fell to about 25–27 °C while simultaneously showing temperature peaks of different magnitude, ranging from 1 °C to as much as 10 °C. The maximum measured temperature (Tmax) was below 40 °C in all measurements. Non-contact Er:YAG laser showed a qualitatively different behavior. Measurements always showed several clearly expressed temperature peaks, about 1 s apart, typically reaching 80–90 °C, with maximum measured temperatures as high as 100 °C. In piezosurgery and low-speed surgical drill measurements similar behavior was observed, but without pronounced temperature peaks (only slight temperature fluctuations were observed).

Measurements of Tbase, Tmax, and ΔT are shown as histograms in Figure 2, Figure 3, Figure 4 and Figure 5. Significant reduction of heat generation after contact Er:YAG laser, piezosurgery, and surgical drill was observed, when compared to non-contact Er:YAG laser osteotomy (Figure 6).

## 4. Discussion

The present in vivo experimental animal study compared the extents and rates of temperature rise following rat tibiae osteotomies with four preparation methods for bone osteotomies: contact Er:YAG laser, digitally controlled non-contact Er:YAG laser, piezosurgery, and low-speed surgical drill. The thermal alterations of the bone tissue produced by the Er:YAG laser irradiation used in contact mode and piezosurgery were minimal as registered by means of an infrared thermographic camera. Digitally controlled non-contact Er:YAG laser produced a temperature that is higher than the recommended border threshold for bone tissue during osteotomy [17], but this temperature peak lasted only for a short time without irreversible changes of bone.

The definition of heat is a process of flowing energy from hot to cold objects. Although it sounds simple, analyzing heat transfer is a complex physical problem. Many research studies have been expanded to measure heat production while bone cutting using a variety of techniques. When dealing with temperature recording in bone tissue there are important factors to consider: pointing the measuring device, the cooling system, the distance from the heat source, and the thermal properties of the bone sample [18].

In previous in vitro study, Pandurić et al. [8] compared an Er:YAG laser (pulse energy, 1000 mJ; pulse duration, 300 μs; frequency, 20 Hz) and surgical drill for osteotomy by assessing the temperature rise. There was a significant statistical difference between the Er:YAG laser and the surgical pilot drill for all measured parameters except the temperature interval. The maximum temperature interval was directly related to the bone thickness and initial temperature of the surface, while in the case of the surgical drill there were no relations between these parameters. In the in vitro study on mandibular bones, Kimura et al. [19] stated that a temperature rise over 10 °C (ΔTa) could be recorded 30 s after laser application. A presented study temperature rise over 10 °C has not been recorded for contact Er:YAG laser. The maximum temperature (Tmax) was below 40 °C in all measurements while non-contact Er:YAG laser showed qualitatively different behavior. Measurements in the presented study always showed several clearly expressed temperature peaks, about 1 s apart, typically reaching 80–90 °C, with maximum measured temperatures as high as 100 °C in time interval of 5 s, needed to achieve 2 mm deep osteotomy. Pandurić et al. [20] reported that spherical formations on the cortical and the trabecular bone of the laser-ablated surface are mostly a result of rapid increase of the temperature during laser irradiation, and then rapid cooling by the integrated cooling system of the laser. Laser irradiation in non-contact mode produces thermo-mechanical ablation on the surface which depends on the energy delivered during radiation.

The Er:YAG wavelength has a high absorption coefficient in water and hydroxyl ions of hydroxyapatite. Bone or dental tissues absorb almost all the energy delivered, which causes an immediate rise in local temperature [21]. In our study significant differences regarding comparison of the temperature changes of the Er:YAG laser in non-contact mode, contact mode, piezosurgery, and surgical drill were found. This is presumably a result of different mechanisms of working, mechanical versus thermal ablation but at the same time, the pulse energy of the non-contact laser was responsible for the fastest time for osteotomy performance with a relatively minimal change in temperature.

Piezoelectric surgery offers many advantages compared with surgical drill and was also an object of our study. Schütz et al. [22] carried out temperature measurements during piezosurgery under conditions as close as possible to clinical practice in pig jaws heated to body temperature. The osteotomies were performed according to the manufacturer’s recommendations: with intermittent pulling movements and minimal contact pressure. Contact pressure was automatically stopped on the device when exceeding 10 N and at a depth of 3 mm. They used three different piezosurgical saws which led to intraosseous temperature increases for 1 min but overall they were below the threshold of 47 °C. This study [22] also reported that the size of the insert for the osteotomies had a pronounced influence on temperature. Increase in bone temperature was smaller when the smallest insert was used compared with the largest one. There are some reports of increased intraosseous temperature up to more than 100 °C during the use of oscillating bone saws, when the coolant did not penetrate the depth of the tip of insert [23]. Results from the presented study showed that Tbase and Tmax temperature during piezosurgery were much lower than for Schütz et al. [22] and for the border threshold [17]. Although contact pressure was higher, depth, drilling time and surface of the used insert instrument were smaller so the results may be explained by previous findings. These findings are in accordance with results by Delgado-Ruiz et al. [24], who compared the curve of temperature and time when drilling in hard bone with and without irrigation. Tmax (28.72 °C) with irrigation was similar to our Tmax (29.15 °C), and it was reported that the thickness of the cortical bone and coolant influenced the temperature increase.

Instead piezosurgery, increased contact pressure in combination with high rotational speed causes quicker bone removal and decreased working time. A non-traumatic surgical technique is critical when preparing and inserting dental implants into bone tissue. The difference between the use of piezosurgery and a conventional drill is regarding the neo-osteogenesis and inflammatory reaction after implant-site preparation. The piezoelectric device stimulated peri-implant osteogenesis, and a reduction of proinflammatory cytokines [25]. Attanasio et al. [26] reported that after analyzing three different osteotomy techniques in medullary bone, Summers osteotomes and bone compactors helped in achieving better primary stability than twist drills. Laser assisted osteotomy has attracted much attention as a promising technique in implant site preparation, especially because of the improved healing induced by biomodulation [27]. One of the major factors for implant failure is the generated heat during the preparation protocol of the implant site. Eriksson and Albrektsson [17] found that heating the titanium implants in rabbit tibia to temperature above 47 °C for longer than 1 min was enough to cause resorption of 30% of the surrounding bone. This was a slow developing process that lasted for 4 weeks, in which the bone was replaced with fat cells, disturbing implant incorporation. The main role in heat production was the pressure and speed of the surgical drill. A study by Brisman [28] showed that increasing the speed and the load together showed no significant increase of temperature. It was suggested that bone temperature was reduced because of drilling at high speed and with a large load. Sharawy et al. [29] evaluated the heat generated from three different drilling speeds (1225, 1667, and 2500 rpm) and concluded that the maximum speed may decrease the risk of intraosseous damage. Slower drilling speed produces more frictional heat because it demands more drilling time. In our study the drilling speed was an average recommended speed (1200 rpm) for the pilot drill for most of the dental implant systems. On analyzing results published by Marković et al. [30] where temperatures between lateral condensation and bone drilling technique were measured and compared, it is apparent that Tbase and Tmax were higher than in our study for surgical drills, but lower for condensation. Marković et al. [30] concluded that the surgical technique used for implant site preparation is the only significant predictor of changes in bone temperature. Regarding this, the main reason for different results is probably the fact that the depth preparation was deeper than in our study.

The thermal effect on adjacent tissues was a major concern of this study. Precise knowledge concerning the heat generation induced by laser, piezosurgery device, and a surgical drill seems to be the key factor of therapeutic success for osteotomy. Osteotomies made by contact Er:YAG laser, piezosurgery, and surgical drill were beyond this temperature limit, while the non-contact Er:YAG laser achieved a much higher temperature. This result should not be clinically relevant because the temperature peaks lasted less than 1 s, until the preparation site was made over 5 s, and this is not enough to cause irreversible changes in bone healing. This is supported by previous study focused on healing [31]. All these facts together make different drilling protocols necessary with the help of modern bone cutting devices.

After osteotomy is performed, the mechanism of bone repair starts with a hematoma, following a fibrinous blood clot which serves as the medium for repair and cell ingrowth. Bone repair is characterized by forming woven bone which transforms into lamellar bone [11]. A study by Esteves et al. [9] found no differences in bone healing in rat tibiae comparing conventional drill osteotomy with piezosurgery. Furthermore, more active osteogenesis was reported in sites prepared with piezosurgery rather than conventional osteotomy with standard steel bur [2]. The results of one of the previous studies [31], which analyzed healing of bone tissue treated with piezosurgery, Er:YAG laser in contact and non-contact modes using laser profilometry, showed the best result for non-contact digitally controlled mode of laser. Healing after non-contact digitally controlled laser was slightly slower in the first week because of temperature changes, but equal or even faster after three weeks. Laser contact mode and piezosurgery showed almost the same result at the early phase of healing. Contact Er:YAG laser has a strong hemostatic effect and causes decreased blood clot formation in the first week of healing period, but after two weeks of healing almost the same results were found for both laser modes. These results are comparable with Yoshino et al. [32], who compared bone healing after electrosurgery and contact and non-contact Er:YAG laser. They reported better results after using two different modes of laser than electrosurgery.

Within the limitations of this experimental in vivo study, such as lack of histological and immunohistochemical analysis and inability of the thermographic camera to measure temperature changes under the surface of bone, the infrared thermography measurement of temperature changes during four different osteotomies revealed scientifically valuable findings. However, there is a need for more extensive and expanded clinical studies to show if temperature changes could affect the speed and quality of bone healing after osteotomies made by the different surgical techniques.

## 5. Conclusions

The present experimental study showed beneficial effects of the osteotomy performed by Er:YAG laser used in the contact mode of working and piezosurgery, reducing the potential overheating of the bone tissue compared to osteotomy with non-contact Er:YAG laser as determined by means of infrared thermography. The results provided important evidence to support, advance, and expand clinical Er:YAG laser and piezosurgery applications in clinical dentistry as safe and predictable methods and alternatives to classical surgical techniques in the wide field of oral surgery, periodontology, and peri-implant therapy.

## Figures and Tables

**Figure 1 materials-14-03051-f001:**
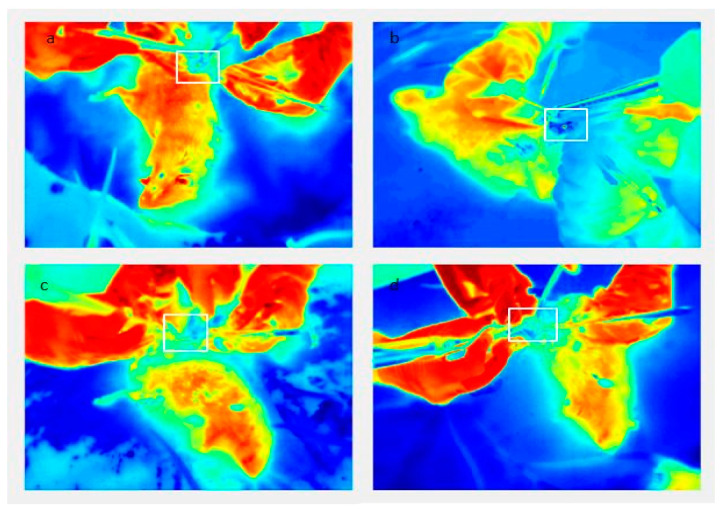
Thermal images with marked region of interest (ROI) for all measured techniques; (**a**) contact Er:YAG laser; (**b**) non-contact Er:YAG laser; (**c**) piezoelectric surgery; (**d**) low-speed surgical drill.

**Figure 2 materials-14-03051-f002:**
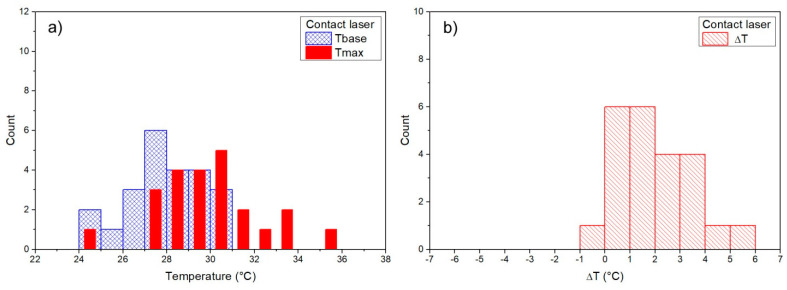
(**a**) Contact Er:YAG laser Tbase and Tmax results (the same binning and axes scale is used to directly compare), (**b**) results for Tmax–Tbase (ΔT).

**Figure 3 materials-14-03051-f003:**
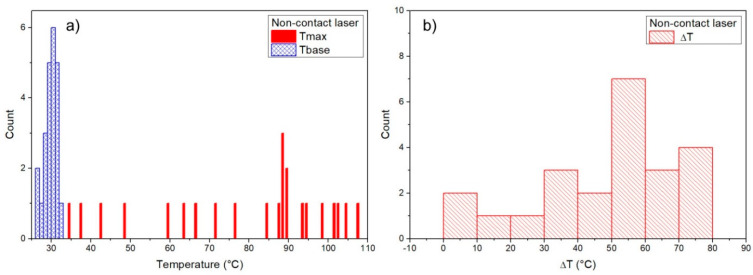
(**a**) Digitally controlled non-contact Er:YAG laser Tbase and Tmax results, (**b**) results for Tmax–Tbase (ΔT).

**Figure 4 materials-14-03051-f004:**
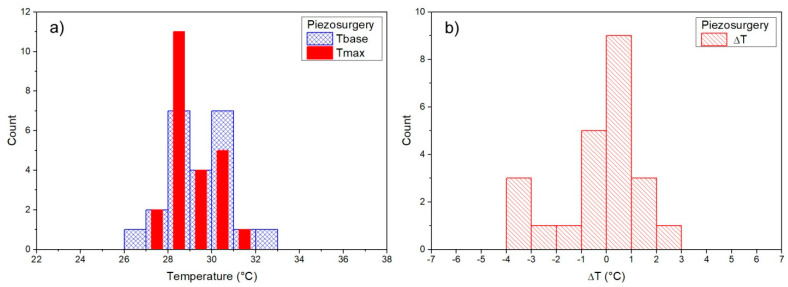
(**a**) Piezosurgery Tbase and Tmax results, (**b**) results for Tmax–Tbase (ΔT).

**Figure 5 materials-14-03051-f005:**
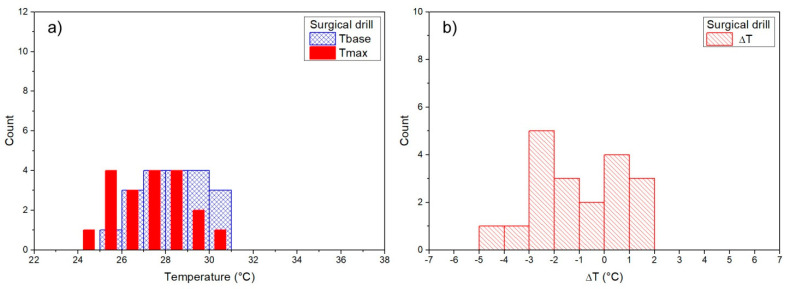
(**a**) Low speed surgical drill Tbase and Tmax results (the same binning and axes scale as in Figure 2 is used to directly compare), (**b**) results for Tmax–Tbase (ΔT).

**Figure 6 materials-14-03051-f006:**
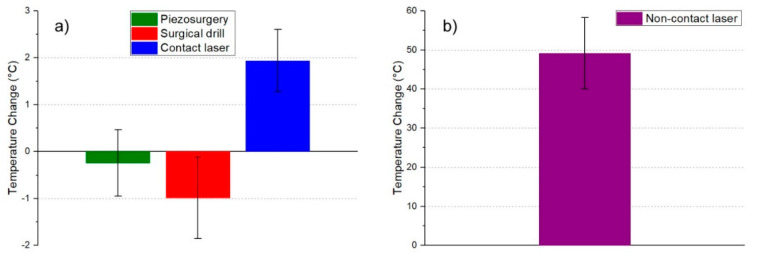
(**a**) Graphic display of temperature changes with marked error bars (shows 95% confidence interval) for piezosurgery, surgical drill and contact laser, (**b**) Non-contact laser was presented on a separate graph for better visibility.

**Table 1 materials-14-03051-t001:** Data normality test (Shapiro–Wilk test).

	N	Mean	SD	W **	*p* *
Contact laser	15	2.00	1.60	0.96	0.63
Non-contact laser	15	53.27	19.15	0.89	0.06
Piezosurgery	15	−0.02	1.62	0.85	0.02
Surgical drill	15	−1.08	1.84	0.95	0.54

* *p*-value for Shapiro–Wilk test, ** W-value of.

**Table 2 materials-14-03051-t002:** Descriptive statistics (sample size, mean, standard deviation, standard error, median, minimum, and maximum) for the four techniques.

								ANOVA
	N	Mean	St.dev.	St.err.	Med.	Min.	Max.	*p*
Contact laser	15	2.00	1.60	0.41	1.87	−0.50	5.08	<0.0001
Non-contact laser	15	53.27	19.15	4.94	59.17	15.80	75.92	
Piezosurgery	15	−0.02	1.62	0.42	0.54	−3.63	2.03	
Surgical drill	15	−1.08	1.84	0.48	−1.25	−4.25	1.89	

**Table 3 materials-14-03051-t003:** Results for mean values of Tbase, Tmax, and ΔT, for each osteotomy technique (Tbase = temperature prior to treatment, Tmax = maximum temperature measured during the treatment).

		Baseline Temp. Tbase (°C)		Maximum Temp. Tmax (°C)		ΔT (°C)	
	N	Mean	St.err.	Mean	St.err.	Mean	St.err.
Contact laser	22	27.98	0.38	29.93	0.53	1.95	0.33
Non-contact laser	22	29.95	0.36	78.88	4.84	48.93	4.60
Piezosurgery	23	29.39	0.28	29.15	0.23	−0.24	0.34
Surgical drill	18	28.35	0.35	27.31	0.39	−1.04	0.43

## Data Availability

The data presented in this study are available on request from the corresponding author.

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
