# Peer review of "Thermal Evaluation by Infrared Thermography Measurement of Osteotomies Performed with Er:YAG Laser, Piezosurgery and Surgical Drill—An Animal Study"

_materials, 2021, doi:10.3390/ma14113051_

Round 1

Reviewer 1 Report

he manuscript submitted to Materials entitled “Thermal Evaluation by Infrared Thermography Measurement of Osteotomies Performed with Er:YAG Laser, Piezosurgery and Surgical Drill – An Animal Study” is a manuscript which aim to investigate the heat development performing osteotomies with Er:YAG Laser, Piezosurgery and Surgical Drill on rat's tibiae.

On my opinion the article is well written. Although it is an animal study on mouse model with poor reference to clinical situations, the content of the manuscript is very interesting with clear results.
Methods are not be even clear.

I highlighted some issues.

-Introduction:

This part introduced the use of these techniques for the bone cutting, however in the discussion part there is a huge part referring implant site preparation. I suggest to improve the introduction also referring to the implant site preparation and the influence of the surgical technique on heath development.

Methods:

Why is there no randomization in the choice of the site to be prepared with the surgical techniques?
This is a bias of the study and I strongly suggest to include it in the study limitation.
In fact there may be a region of the tibia with a greater medullary component (Diaphysis) or cortical (epiphysis).

How did you calculate the pressure applied during the surgical procedures?

In the methods is specified the piezoelectric tip (S2) for the osteotomy.
However, the use of S2 tip is suggested for the coronotomy or to design the bone window for the lateral sinus lift. It is not the piezoelectric insert for the implant site preparation.

Furthermore, although in the introduction part reference is made to the osteotomy technique and not to implant site preparation, the laser, a piezo insert for the lateral sinus lift window and a round steel surgical bur  (without specifying whether diamond or multi-blade) with 2-mm- are used as procedures. 

In my opinion there are a lot of bias in the methods part which can affect the quality of the work.

-Discussion:

"To the best of our knowledge, the comparison of temperature changes of Er:YAG laser, piezosurgery and surgical drill has not been previously discussed in the literature"
Please, delete this sentence. There are many studies in literature that compare heat generation with these procedures.

"Contact pressure was automatically stopped on the device from exceeding 10 N"
How did you measured it? And how did you stopped the device?

"reported that the size of the insert for the osteotomies had a pronounced influence on temperature. Increase in bone temperature was smaller when the  smallest insert was used compared with the largest one."
Did you analyzed different insert size? Please specify

"Non traumatic surgical technique is critical when preparing and inserting dental implants into bone tissue"
I suggest to improve the significance of this sentence with references:
PMID: 26635486 PMID: 32098046 PMID: 27548111 

"
Eriksson and Albrektsson [17] found that threshold level for irreversible bone changes is increase of more than 10 0C in temperature during period longer than 1 minute"
This sentence was cited in a previous part of the discussion, please delete it.

"Within the limitations of this experimental in vivo study, infrared thermography measurement of temperature changes during four different osteotomies revealed scientifically valuable findings."
I strongly suggest to specify the limitations of this study (lack of radiological evaluation, no randomization, improper use of inserts...)

After a careful review of the manuscript and the clarifications requested by the reviewers, the article can be re-evaluated for possible publication.

Author Response

Dear Reviewer,

After receiving the revision of our manuscript, we thank You for constructive comments and suggestions. We agree with the majority of your suggestions so we have revised the manuscript accordingly. All changes in the revised manuscript have been made and highlighted in the text and reponses to Your comments are given below.

Reviewer's report:

The manuscript submitted to Materials entitled “Thermal Evaluation by Infrared Thermography Measurement of Osteotomies Performed with Er:YAG Laser, Piezosurgery and Surgical Drill – An Animal Study” is a manuscript which aim to investigate the heat development performing osteotomies with Er:YAG Laser, Piezosurgery and Surgical Drill on rat's tibiae.

On my opinion the article is well written. Although it is an animal study on mouse model with poor reference to clinical situations, the content of the manuscript is very interesting with clear results.
Methods are not be even clear.

I highlighted some issues.

Comment 1:

Introduction: This part introduced the use of these techniques for the bone cutting, however in the discussion part there is a huge part referring implant site preparation. I suggest to improve the introduction also referring to the implant site preparation and the influence of the surgical technique on heath development.

Response: This part was added in the introduction part.

Comment 2:

Methods: Why is there no randomization in the choice of the site to be prepared with the surgical techniques?
This is a bias of the study and I strongly suggest to include it in the study limitation.
In fact there may be a region of the tibia with a greater medullary component (Diaphysis) or cortical (epiphysis).

Response:  Thank you for your comments. Here is the explanation. Bone thickness was one of the most important factors that was considered in designing the study. The tibia has two parts: the diaphysis and the epiphysis. The diaphysis is the tubular shaft that runs between the proximal and distal ends of the tibia. The walls of the diaphysis are composed of dense and hard compact bone. Hence, the basic approach in the drilling of osteotomies was to localize them only in the diaphysis. Also during drilling, exceptional attention was made to the deep of the osteotomies. Actually, in all animals osteotomies were localized in dense and hard compact bone to ensure uniformity for all osteotomies performed.

On the other hand the wider section at each end of the tibia is called the epiphysis which is filled with spongy bone. This was the second reason why osteotomies were localized in diaphysis because drilling in the epiphysis could reach the spongy bone and induce different mechanisms not typical for hard compact bone. Tibia and rat calvaria are common models for exploring bone mechanisms. In this study, we selected the tibia because of the  adequate osteotomy depth, since in the case of calvaria model, the osteotomy depth would reach to the dura.

Comment 3: How did you calculate the pressure applied during the surgical procedures?

Response: During the preparations, an articulated arm delivery system of the laser, piezoelectric and low-speed handpieces were fixed with same pressure in all cases. We rewrote the sentence in the text to make it clear.

Comment 4: In the methods is specified the piezoelectric tip (S2) for the osteotomy. However, the use of S2 tip is suggested for the coronotomy or to design the bone window for the lateral sinus lift. It is not the piezoelectric insert for the implant site preparation.

Response: We were aware of this fact, but in this study S2 tip was used becuase of technical reasons. Rat's tibia is very small and its width is a little bit wider than S2 tip, so usage of other piezolectric's tips was technically  impossible. Due to the fact that lateral bone window preparation is also a type of osteotomy, we chose the above tip. It is also important to note that most piezoelectric systems do not have factory-made tips for implant site preparation.

Comment 5: Furthermore, although in the introduction part reference is made to the osteotomy technique and not to implant site preparation, the laser, a piezo insert for the lateral sinus lift window and a round steel surgical bur  (without specifying whether diamond or multi-blade) with 2-mm- are used as procedures. 
In my opinion there are a lot of bias in the methods part which can affect the quality of the work.

Response: Round surgical bur which was used in this study is one that is usually use in clinical cases and studies for osteotomy and that was a multi-blade type of round bur. That was the reason why we did not emphasize this fact. Preparation of dental implant bed is also type of bone osteotomy. All osteotomies were identical, 2 mm deep and with the same distance between them. The distance of 5 mm between two different types of osteotomies were selected to ensure that one technique would not affect the change in bone temperature at the site of application of the other technique. Basic idea was that all osteotomies should be exactly the same, of the same diameter and the same depth after preparation in order for the comparative results to be as accurate as possible.

Comment 6: Discussion: "To the best of our knowledge, the comparison of temperature changes of Er:YAG laser, piezosurgery and surgical drill has not been previously discussed in the literature"
Please, delete this sentence. There are many studies in literature that compare heat generation with these procedures.

Response: According to our best knowledge, there are no studies that compare heat generation with this type of thermographic procedure and by the means of infrared thermographic camera. Since we did not clearly write the method used in this sentence and the meaning can be understood in several ways, this sentence was deleted.

Comment 7: "Contact pressure was automatically stopped on the device from exceeding 10 N"
How did you measured it? And how did you stopped the device?

Response: This sentence was referred to the reference No. 22 and does not relate to the methodology of our research.

Comment 8: "reported that the size of the insert for the osteotomies had a pronounced influence on temperature. Increase in bone temperature was smaller when the  smallest insert was used compared with the largest one."
Did you analyzed different insert size? Please specify

Response: This was also referred to the findings published by: Schütz, S.; Egger, J.; Kühl, S.; Filippi, A.; Lambrecht, J.T. Intraosseous temperature changes during the use of piezosurgical inserts in vitro. Int J Oral Maxillofac Surg. 2012, 41, 1338-1343, doi: 10.1016/j.ijom.2012.06.007., and does not relate to the results of our research.

Comment 9: "Non traumatic surgical technique is critical when preparing and inserting dental implants into bone tissue" I suggest to improve the significance of this sentence with references:
PMID: 26635486 PMID: 32098046 PMID: 27548111 

Response: This part was extended with text and references were added.

Comment 10: "Eriksson and Albrektsson [17] found that threshold level for irreversible bone changes is increase of more than 10 0C in temperature during period longer than 1 minute"
This sentence was cited in a previous part of the discussion, please delete it.

Response: Sentence was deleted as suggested by the reviewer.

Comment 11: "Within the limitations of this experimental in vivo study, infrared thermography measurement of temperature changes during four different osteotomies revealed scientifically valuable findings."
I strongly suggest to specify the limitations of this study (lack of radiological evaluation, no randomization, improper use of inserts...)

Response: Limitations of this study are lack of histological analysis and inability of infrared thermographic camera to measure temperature changes in the depth of the bone and under the surface of bone. Histological and histomorphomethrical analysis of osteotomies prepared by different preparation techniques are planned to do in the next animal study as a continuation of this research method to evaluate the healing process related to different osteotomy techniques and temperature changes.

The limitations of present study are added in the main text, as suggested.

Comment 12: After a careful review of the manuscript and the clarifications requested by the reviewers, the article can be re-evaluated for possible publication.

Response: Thank you very much for your valuable comments and suggestions,  and we did our best to make all requested changes in the manuscript.

Reviewer 2 Report

Dear Authors, below are my comments about the submitted manuscript.

  1. I am skeptical about way the main topic of the study can comply with the Journal’s aims and scopes. There are several Journals with higher specificity.
  2. The title of the manuscript well conveys with the major concern of the study.
  3. The abstract is well structured and properly summarize the topic addressed.
  4. The references are up to date.
  5. The Introduction section well sets up the main topic and introduce the development of the manuscript. However, the null hypothesis is not expressed. The null hypothesis statement must precisely identify the variables assessed through statistical analysis. Please add its statement in the last part of the Introduction section, following the aims of the research.
  6. Why did you not perform the sample size calculation? Did you base on previously published study? Please clarify. Furthermore, I do not get the choice to arrange the statistical analysis on the Student T test for the difference detection between groups. Why did you not perform an ANOVA test with post hoc test, for the analysis of differences within the groups and among the groups? It should not be more meaningful? Please clarify. Clarify also which are the statistical units considered.
  7. I retain you should do a better job in displaying and explain Figure 2 to 5.
  8. I think the study will gain more impact if a deeper statistical investigation and bioptic harvesting with histological analysis will add.
  9. Discussion section is too long with respect to the other sections.
  10. Which can be the clinical implication of your result? Please clarify.

Author Response

Dear Reviewer,

After receiving the revision of our manuscript, we thank You for constructive comments and suggestions. We agree with the majority of your suggestions so we have revised the manuscript accordingly. All changes in the revised manuscript have been made and highlighted in the text and reponses to Your comments are given below.

Dear Authors, below are my comments about the submitted manuscript.

Comment 1: I am skeptical about way the main topic of the study can comply with the Journal’s aims and scopes. There are several Journals with higher specificity.

Response: Thank you for your comment, after an editor's invitation for paper for this issue, we sent an abstract which was evaluated and approved by the Editor. After receiving the Editor's proof based on the abstract of the study we have submitted the full version of the manuscript.

Comment 2: The title of the manuscript well conveys with the major concern of the study.

Response: Thank You.

Comment 3:The abstract is well structured and properly summarize the topic addressed.

Response: Results in teh abstract part were re-written and corrected according to the new statistical analysis.

Comment 4: The references are up to date.

Response: Thank You, three references were added according to the reviewer's suggestion.

Comment 5: The Introduction section well sets up the main topic and introduce the development of the manuscript. However, the null hypothesis is not expressed. The null hypothesis statement must precisely identify the variables assessed through statistical analysis. Please add its statement in the last part of the Introduction section, following the aims of the research.

Response: Thank you for the suggestion. Null hypothesis is clearly expressed and added at the end of the introduction part.

Comment 6:Why did you not perform the sample size calculation? Did you base on previously published study? Please clarify. Furthermore, I do not get the choice to arrange the statistical analysis on the Student T test for the difference detection between groups. Why did you not perform an ANOVA test with post hoc test, for the analysis of differences within the groups and among the groups? It should not be more meaningful? Please clarify. Clarify also which are the statistical units considered.

Response: Thank you very much for this observation. According to your recommendations, the completely new statistical analysis was performed. This was the main reason why we kindly asked the Editor to extend the deadline for returning the corrected manuscript. Sample size calculation was added and highlighted in the text. Results were re-written accordingly, as well as structured abstract.

Comment 7: I retain you should do a better job in displaying and explain Figure 2 to 5.

Response: We agree with your opinion and suggestion, the added part of the text is highlighted in the yellow. Additionally, new tables were made and added in the section results.

Comment 8: I think the study will gain more impact if a deeper statistical investigation and bioptic harvesting with histological analysis will add.

Response: Thank you for the comments. As suggested, deeper statistical analysis was performed. We strongly agree that histological and immunohistochemical analysis would be interesting for the scientific community and future clinical implications so we are planning another experimental research focused on bone healing related to temperature changes produced using different osteotomy techniques.

Comment 9: Discussion section is too long with respect to the other sections.

Response: Other sections were expanded, introduction part as suggested by the reviewer, methods by the new statistical analysis, results section changed and re-written according to the new analysis. The conclusion part was re-written, as suggested.

Comment 10: Which can be the clinical implication of your result? Please clarify.

Response: The results from this study provided an important evidence to support, advance and expand clinical applications contact and non-contact mode of Er:YAG laser and piezosurgery in the clinical dentistry as a safe and predictible method and alternative to classical surgical techniques in the wide field of oral surgery, periodontology and peri-implant therapy.

Since the temperature changes that occur during the laser and piezosurgical bone preparation should not affect further bone healing, these techniques can be applied in everyday clinical practice. Of course, further studies related to the bone healing based on histology, immunohistochemistry and histomorphometry are needed.

Reviewer 3 Report

Line 131-132 - pulse energy listed is 375 mJ, when the manufacturer recommends 500 mJ. Please check.

Table 1 . you should check the results since it seems that piezosurgery and surgical drill resulted in lower working temperatures than baseline. Is that correct? Plus you did not include any detail about the statistical significance in the Results section, but you mention those in Discussion section.

Additionally, X-Runner is not created and has no indications for clinical use in osteotomy - you should mention that. It is simply too aggressive.

Line 246 - I suggest you delete this statement because you discuss it only later on.

You should re-write Conclusions because they do not reflect what you have in Results.

Author Response

Dear Reviewer,

After receiving the revision of our manuscript, we thank You for constructive comments and suggestions. We agree with the majority of your suggestions so we have revised the manuscript accordingly. All changes in the revised manuscript have been made and highlighted in the text and reponses to Your comments are given below.

Comment 1: Line 131-132 - pulse energy listed is 375 mJ, when the manufacturer recommends 500 mJ. Please check.

Response: According to the results of the previous pilot study on the optimal contact Er:YAG laser irradiation parameters on bone tissue healing, we set parameters for this study as: power 7.5 W, pulse energy 375 mJ, repetition rate 20 Hz, and medium-short pulse mode (MSP mode, pulse duration 10 μs). The experimental study protocol (pilot and final) before starting the research was sent to the manufacturer for checking selected parameters (Fotona d.d., Ljubljana, Slovenia). The scientific committee in the manufacturer's lab has corrected and approved the protocol of using contact and non-contact mode of Er:YAG laser for rat tibia osteotomies.

Comment 2: Table 1 . you should check the results since it seems that piezosurgery and surgical drill resulted in lower working temperatures than baseline. Is that correct? Plus you did not include any detail about the statistical significance in the Results section, but you mention those in Discussion section.

Response: A new statistical analysis was made and added in the manuscript and this was corrected. We have repeated all thermographic readings results for all animals and all preparations. Only temperature results that could be clearly read from all measurement points in the area of ROI were used. Values of the temperature at which the instrument holder or fixative has partially covered the surface of the bone on which the value was measured were excluded from the sample in the new statistical analysis. This explanation was added in the methodology part to make it clearer to the reader, thank you for your suggestion.

Comment 3: Additionally, X-Runner is not created and has no indications for clinical use in osteotomy - you should mention that. It is simply too aggressive.

Response: Thank you for this comment. According to the official web site of manufacturer (Fotona d.d., Slovenia), it has been strongly indicated for clinical use in surgical and dental implantology  procedures. The main benefit of it's usage in surgical techniques is high bactericidal and detoxification effects, absence of metal abrasion, reduced tissue bleeding and lack of vibration during procedures. https://www.fotona.com/media/documents/92943_v1_x_runner_sx02_leaflet_16.pdf

Comment 4: Line 246 - I suggest you delete this statement because you discuss it only later on.

Response: This statement was deleted, as suggested by the reviewer.

Comment 5: You should re-write Conclusions because they do not reflect what you have in Results.

Response: Conclusion was corrected and rewritten, according to the reviewer's comments.

Round 2

Reviewer 1 Report

Although many critical issues remain in the paper, the authors cannot make further changes to improve the quality of the manuscript

Reviewer 2 Report

Well done